# “Omics” Techniques Used in Marine Biofouling Studies

**DOI:** 10.3390/ijms241310518

**Published:** 2023-06-23

**Authors:** Sergey Dobretsov, Daniel Rittschof

**Affiliations:** 1Department of Marine Science and Fisheries, College of Agricultural and Marine Sciences, Sultan Qaboos University, Al Khoud 123, Muscat P.O. Box 34, Oman; 2Nicholas School of the Environment, Duke University, Beaufort, NC 28516, USA; ritt@duke.edu

**Keywords:** fouling, antifouling, metagenomic, metabolomic, transcriptomic, proteomic

## Abstract

Biofouling is the growth of organisms on wet surfaces. Biofouling includes micro- (bacteria and unicellular algae) and macrofouling (mussels, barnacles, tube worms, bryozoans, etc.) and is a major problem for industries. However, the settlement and growth of some biofouling species, like oysters and corals, can be desirable. Thus, it is important to understand the process of biofouling in detail. Modern “omic” techniques, such as metabolomics, metagenomics, transcriptomics, and proteomics, provide unique opportunities to study biofouling organisms and communities and investigate their metabolites and environmental interactions. In this review, we analyze the recent publications that employ metagenomic, metabolomic, and proteomic techniques for the investigation of biofouling and biofouling organisms. Specific emphasis is given to metagenomics, proteomics and publications using combinations of different “omics” techniques. Finally, this review presents the future outlook for the use of “omics” techniques in marine biofouling studies. Like all trans-disciplinary research, environmental “omics” is in its infancy and will advance rapidly as researchers develop the necessary expertise, theory, and technology.

## 1. Introduction

### 1.1. Biofouling

Organisms living on surfaces take advantage of the energetics of water movement to carry away wastes and propagules and obtain food. The term biofouling is usually used for the growth of organisms on manufactured surfaces [1]. Biofouling includes microorganisms (microfouling) and macroscopic organisms (macrofouling) (Figure 1). Biofilms include layers of microorganisms in a matrix of organic molecules attached to surfaces [2].

Microfouling includes bacteria, archaea, fungi, microalgae, sessile metazoans, and microalgae [3]. Microfouling includes microbes living in biofilms attached to a hard substratum, the water’s surface, marine snow, and other particles [4]. Microfouling results from the active or passive movement of microbes onto the surfaces from the water column. The chemical and physical properties of the substratum as well as the presence of other microbes determine the adhesion and biofilm formation of different bacteria. Microfouling can be instantaneous or take several hours for the transition of individual bacteria from the water column to a sessile existence [5]. Microfouling is dynamic and much of it can be reversed.

Macrofouling includes metazoan organisms with at least one sessile life stage (Figure 1). As in the case of microfouling, macrofouling can be immediate upon submergence of a surface or after a while when the biofilm is transformed [6]. Usually, macrofouling is due to attachment to a surface by a propagule/dispersal life stage. Upon attachment, propagules metamorphose into juvenile organisms [7]. Attachment can be instantaneous upon contact with the surface as in the case of many ascidian larvae. Alternatively, attachment can be longer; minutes for bryozoans and barnacles and even longer as is the case for tubeworm larvae which require pre-existing biofilms. Thus, macrofouling processes can take a few minutes to a few days. Some propagules require biofilms for their settlement and metamorphosis [7,8,9]. Many macrofoulers are gregarious and gregariousness is mediated by cues and pheromones [6]. Cues can originate from various sources including symbiotic microalgae, conspecifics, generated by exoproteases, molecules leaching from degrading biological adhesives, and anthropogenically derived synthetic mimics. Most macrofouling is not reversible.

Most sessile macrofoulers have planktonic larval stages that function in the dispersal and colonization of new substrates. These larval stages are complex in their physical, behavioral, and chemical response repertoire and have entertained biologists for tens of decades. Larval settlement is a logical control step for biofouling management [8].

### 1.2. Antifouling Techniques

Organisms living on manufactured surfaces degrade performance. Degradation is by corrosion, the build-up of biomass that causes resistance to flow decreasing structural stability and increasing resistance [10]. On hulls, speed, and fuel consumption are compromised. Biofouling costs money. There are over 2000 years of history of attempts to control biofouling [11].

Historically, biofouling is mediated by killing what grows on surfaces [11]. Commercial solutions are broad-spectrum biocides ranging from chlorine gas which oxidizes organisms to heavy metal ions which disrupt metabolic processes [10,12]. As environmental damage is recognized, biocide regulations are increased and the most toxic ones are banned. In recent decades, live organic biocides have been added to the commercial repertoire. These co-biocides can be more environmentally damaging than the biocides they augment. A recent twist on using toxic metals is nanomaterials like nano-silver, nano-copper, nano-titania, and nano-zinc [13,14,15]. Nanomaterials generate reactive oxygen species and release toxic metal ions that kill biofouling organisms [16].

Coatings companies are now developing foul-release technology, a major improvement [12,17]. This technology is proposed as non-toxic and environmentally benign. However, due to loopholes in the regulations, these coatings routinely contain toxic compounds that kill organisms and alter their enzymes and adhesives [18]. Thus, new and non-toxic antifouling solutions are still a quest.

### 1.3. Enhancement of Biofouling

Biofouling has not only negative but also positive consequences. Coral reefs are the largest fouling communities in the world, occupying an area as large as 284,000 km^2^ and is a habitat for many other marine species [3]. Mussel beds and oyster reefs are other examples of large fouling communities [19].

Due to anthropogenic effects, coral reefs and marine ecosystems are under constant threat. However, restoration of damaged ecosystems, like coral and oyster reefs, is possible by enhancement of settlement of larvae of these species onto new substrata [19]. Larvae of corals and oysters require specific chemical cues/pheromones that can be used in order to trigger the release of gametes and larval settlement in the laboratory [20,21,22,23,24]. Later the substrates with young settlers can be moved to the desired habitats. The manipulation of spawning and larval settlement of endangered biofouling species requires extensive knowledge of their chemical ecology and species–species interactions.

Some biofouling species are cultivated and used for food production. These include mussel species like *Mytilus, Saccrostrea* and *Perna.* These species use chemical and biological cues to find suitable substratum and for metamorphosis [7,9,20,21]. To successfully cultivate them, one needs to know how to deliver the chemical cues involved in the process and mechanisms of larval settlement and metamorphosis.

### 1.4. Omics and Biofouling Studies

Traditionally, biofouling research focused on isolating and identifying single micro- and macrofouling organisms and investigating their biological properties (attachment, settlement behavior, metamorphosis, etc.) and chemical metabolites. However, recently so-called “omics” approaches have been used to study complex biofouling communities and their environmental interactions. These “omics” methods include metagenomics, metabolomics, transcriptomics, and proteomics approaches (Table 1, Figure 2). The main questions that these novel approaches try to answer are “who is there?”, “what are they doing?”, “which compounds (proteins) are produced?”, and “how does the environment affect biofouling organisms and their metabolites?”.

Data on the percentage of biofouling studies that use at least one “omics” technique remains low (Figure 3). Since 2015, the percentage of publications has increased from 15% to 23%. Because the figure overcounts studies that use multiple omics approaches, the actual percentage is lower. Most biofouling studies use proteomics to investigate the synthesis of proteins of biofouling organisms and to study their adhesive proteins (Table 1). The second most common type of publication uses metagenomic and transcriptomic approaches (Figure 3). Transcriptomics is used to investigate genes that are transcribed during biofilm formation or larval settlement and metamorphosis [7,25]. Most metagenomic studies use 16S or 18S ribosomal RNA to identify micro- and macroorganisms in biofouling communities (Table 1). However, there are some studies that investigate genes present in biofilms and larvae of macrofouling organisms. Studies investigating metabolomes of biofouling organisms or complex communities are rare but their percentage increases yearly (Figure 3). In particular, biofilms associated with corals [26,27], seaweeds [28,29], sponges [30,31], and other marine organisms [32] are reported.

Several reviews about “omics”-based studies have been published. The use of “omics” datasets for the identification of marine algal natural products is reviewed [33] and the role of “omics” in the exploration of marine phytoplankton is evaluated [34]. A recent review investigates the use of “omics” tools for assessing the biodeterioration of cultural heritage artifacts [35]. The role of “omics” approaches in studying proteins of invertebrates was also reviewed [36]. “Omics” research of an important aquaculture species, like abalone, is summarized [37]. Other reviews investigate the role of biofilms in biofouling [38] and the induction of larval settlement by marine biofilms [7]. However, “omics” approaches in biofouling studies have not been reviewed.

The main aim of this review is to evaluate the use of different omic techniques in biofouling research. In particular, we review biofouling-related metagenomic, transcriptomic, proteomic, and metabolomic studies. We provide examples of marine biofilms on manufactured surfaces and the settlement of commercially important invertebrate species and those that cause significant industrial damage. Studies of biofouling on living organisms (epibiosis) are omitted. Specific emphasis is given to metagenomic and proteomic studies. Finally, we highlight the future directions of using omic techniques and a combination of omic techniques in biofouling studies.
ijms-24-10518-t001_Table 1Table 1The “omics’ approaches used in biofouling studies.Omics ApproachMethodsProposeApplicationsSelected ReferencesMetagenomicsDNA sequencingThe genome of biofouling organisms and their genesIdentification of micro- and macroorganisms in complex communities[39,40,41,42,43]Identification of genes of biofouling organismsTranscriptomicsRNA sequencingTranscripts and their functionsAnalysis of the activity of organisms, and phenotype analysis[44,45,46]MetabolomicsIdentification of compounds via MS (LC-MS/MS, GC-MS)Analysis of all metabolitesMetabolites and pathways of a single biofouling organism or complex communities[47,48]ProteomicsIdentification of proteins via MALDI-TOF and X-ray crystallographyAnalysis of proteinsProteins, enzymes, settlement cues, and glues of biofouling organisms[49]MS—mass spectrometry including LC-MS/MS (liquid chromatography) and GC-MS (gas chromatography). MALDI-TOF—matrix-assisted laser desorption ionization-time of flight mass spectrometry.


## 2. Omics in Context

“Omics” techniques can be daunting for beginners. These techniques originate in the health sector and are dominated by for-profit businesses, proprietary kits, platforms, and special language for each subdiscipline that includes acronyms and jargon. If you were involved in the initial planning and funding the development of “omics”, you would know that there were many assumptions and limits placed on funding for this technology. An example is funding and planning for initial work usually stopped at the cellular level. Today, one can see the consequences of this narrow approach as the use of this technology spreads to human health, microbial ecology, metazoans, biofouling and other communities and environmental health.

At the hands-on level, the kits and specialized tools make benchwork easy. A drawback to the kits approach is that many users do not understand what they are doing. Lack of understanding makes it hard to solve problems and to understand what kind of kits might be correct for the interest you have. If ones work does not fit exactly with the design of kits, then there may be nuances that impact the research.

Finally, “omics projects” generate what is known as “big data”. It includes such vast amount of data that new data analysis is essential and comes with its own jargon, like pipeline and trivial names for analysis techniques. Finally, the data are analyzed using different statistical methods. For most users, the solution is to take advantage of the services provided by centers at universities or private companies that accept samples at some level, process them to generate the data, and then do preliminary analysis that enables a user to answer their specific questions.

To summarize, beginning “omics research” is like going to a foreign country without being able to read or speak the language. Luckily, the “omics” centers are jammed with kind and helpful people who will help you with the basics. However, these people do not necessarily know about marine biology and biofouling. Thus, they help you learn the language, the culture, the power, and the limits of “omics” technology. Once you can understand, exciting new pathways for answering your questions will emerge. As a user, you can tweak what you are doing to help you understand the output and answer your specific research questions.

## 3. Use of Metagenomics in Biofouling Research

Pioneering studies of Prof. Claude E. ZoBell and his colleagues from the Scripps Institution of Oceanography (USA) provided the first information about the diversity of marine microbes in marine biofilms. Culture-dependent techniques were developed that allowed researchers to isolate and investigate the properties of marine bacteria for years. However, less than 1% of marine bacteria can be cultivated in the laboratory using traditional techniques [50]. Thus, culture-dependent techniques limit our understanding of microbial diversity. Modern metagenomic techniques allow the identification of all microbes from biofilms and seawater and estimate their abundances (Table 2).

The majority of metagenomic studies use amplicon sequencing of small pieces (300–1000 bp) of 16S ribosomal DNA for prokaryotes and pieces of 18S ribosomal DNA (300 plus bp) and pieces of COI (about 800 bp, cytochrome oxidase I) for eukaryote identification in biofouling communities. Metagenomics is the study of DNA sequences from microbes and metazoans [51]. A key step is the use of the polymerase chain reaction (PCR) for making many copies of existing sequences of DNA that have been extracted from your environmental sample. For PCR, you use specific DNA “primer” sequences that you add that enable you to amplify specific genes [52]. The amplified DNA is called amplicons. You use amplicons to identify organisms (microbes, animals, or plants) by comparison of the obtained DNA sequences with ones in the database. There are many databases but the most used ones are NCBI (National Institutes of Health, USA) and RDP (Ribosomal Database Project from Michigan State University, USA).

The first marine metagenomic study was conducted in 1991 by Dr. Norman R. Pace and colleagues who cloned 16S DNA of the picoplankton community from the Pacific Ocean [53]. A few years later, one of the first metagenomics studies of biofilms was conducted on sulfate-reducing bacteria from marine sediments [54]. In this study, 16S DNA genes were cloned in *E. coli* and identified using Sanger sequencing. The development of sequencing techniques (Table 2) and the reduction in costs resulted in many biofouling studies that utilize metagenomic approaches, which are reviewed using selected examples below.
ijms-24-10518-t002_Table 2Table 2Summary of sequencing platforms used in metagenomic studies.Sequencing GenerationTools UsedFeaturesPropose of StudyExample of Publications1st sequencing generationSanger sequencingUses capillary electrophoresisSequencing of genes; Identification of single biofouling organisms;Full genome sequencing[54,55,56]2nd sequencing generationPyrosequencingMiSeq; HiSeq; Ion TorrentUses labeled nucleotides or detection of hydrogen or lightIdentification of microbes in biofilms; Identification of genes[39,40,41,42,43]3rd sequencing generationOxford NanoporeNo need for PCR amplificationFull genome sequencing; Identification of microbes in biofilms[57]Pyrosequencing—454 Roche pyrosequencing; MiSeq, HiSeq—Illumina platform; Ion Torrent—Ion Torrent semiconductor sequencing technology; Oxford Nanopore—MinION Oxford Nanopore Technologies.


### 3.1. Metagenomics of Biofilms on Man-Made Substrata

Biofilms are a significant part of biofouling. Knowing the composition of the biofilm community is important for environmental toxicology, forensics, understanding surface microbe interaction, management, and ecology. For example, investigators using high throughput 454 pyrosequencing, based upon the detection of light released in the time of nucleotide incorporation during the polymerase chain reaction [58] of 16S DNA genes. Analysis of pyrosequencing results demonstrated that bacterial communities developed on black and white panels exposed to fouling in the sea were different [39]. However, classes Alphaproteobacteria and Firmicutes dominated in all biofilms. Another study that used the same next-generation sequencing (NGS) technique investigated the composition of microbial communities developed in a bioreactor [59]. The investigators found that microbial diversity decreased under high aeration. Similarly, results of pyrosequencing demonstrate that aeration affected the diversity and species richness of bacterial and archaeal communities on osmosis membranes [40].

Illumina is next-generation, high-throughput “sequencing by synthesis technology” based on tracking the addition of fluorescently labeled nucleotides during DNA polymerization [58]. Illumina technology has frequently been used in biofouling metagenomic studies for the analysis of species composition of microbial biofilms developed on man-made structures [60,61,62]. Analysis of microbial communities developed in membrane bioreactors via Illumina sequencing showed that high salinity increased the proportion of *Flavobacterium*, *Aequorivita*, *Gelidibacter*, *Microbacterium*, and *Algoriphagus* genera [41]. Another study investigated biofouling developed on sea gliders using 16S and 18S amplicon sequencing [63]. The researchers observed differences in the number of OTUs (operational taxonomic units) between biofilms on different parts of the glider. Bacteria belonging to the classes gamma- and alpha-proteobacteria dominated prokaryotic communities, while hydrozoans and Chlorophyta dominated eukaryotic communities. Distinct bacterial communities were detected using MiSeq Illumina on stainless steel exposed to biofouling in a Northern Portugal port [64]. In contrast, bacterial communities developed on stainless steel, polyethylene and titanium investigated using MiSeq Illumina shared some similarities [65]. However, the communities changed over time. Most studies using NGS technology report that microbial communities developed on submerged surfaces differ from those in seawater [63,66].

High throughput sequencing using the Illumina platform is used in studies of metabolic assembled genomes from biofouling communities. For example, a study by Walter et al. [67] of microbial mats of a coastal lagoon showed that they are dominated by cyanobacteria responsible for photosynthesis, Chroococcales responsible for nitrogen and ammonia assimilation, and Desulfobacterales contributing to sulfate reduction. Taxonomic and functional metagenomic analysis of biofilms developed in different locations and their effect on larval settlement of the polychaete, *Hydroides elegans,* was investigated [68]. The investigators demonstrated that the microbial communities were significantly different in coastal waters as compared to off-shore waters. However, the functional genes were similar between sites and related to carbohydrates, amino acids, and protein metabolism.

The temporal shift of microbial communities on wood and concrete used for artificial reefs was investigated using Ion Torrent sequencing technology [69]. This technology is based on detecting hydrogen ions during DNA polymerization [70]. The investigators found that the relative abundances of bacterial phyla decreased differently on different substrata over time [69]. Similarly, microbial communities developed in brackish waters on different classes of stainless steel showed that one type of steel had different microbial communities dominated by Actinobacteria, while Proteobacteria dominated other types. Ion Torrent sequencing technology is also used to examine microorganisms on membranes [42] and in bioreactors [71].

MinION Oxford nanopore is a new sequencing technology (Table 2). It can be used for amplicon sequencing as well as for the assembly of full genomes. The advantage of this technique is that it does not require a PCR step and can detect femtograms of DNA [72]. It may be so that because this technology is novel, only a few biofouling-related studies were found (Table 2). Complete genomic sequences of biofouling bacteria *Vibrio cambelli* [57,73] and *Pseudomonas putida* [74] were determined using this technology. Similarly, the genome of important fouling species was sequenced using Illumina and Oxford nanopore technologies [43].

NGS technology can be used to assess the composition of biofilms and investigate their effect on the settlement of macrofoulers. Biofilms and their mussel-inducing activity were investigated via Illumina MiSeq [75]. The phylum Proteobacteria dominated all biofilms. The composition of biofilms inducing *Mytilus coruscus* settlement was different from low inductive ones. The coral settlement-inducing activity of biofilms of crustose coralline alga was investigated using Illumina metagenomics and isolation of bacteria [76]. Data analysis revealed no correlation between inductive settlement capacities and species of bacteria. A recent study demonstrated that biofilms formed in different environmental conditions affect the formation of macrofouling communities [77]. Biofilms were developed in areas of high and low anthropogenic impact and then were translocated. A low settlement rate of non-indigenous species *Watersipora subatra* on biofilms developed in marine protected areas and moved to the area with high anthropogenic impact. This demonstrates that metagenomics can be a useful tool in marine conservation.

Many studies investigated biofouling communities developed on plastics floating in the oceans or deposited in sediments using metagenomic approaches. Due to limited space, we refer readers to reviews about this topic [78,79,80]. In general, the studies confirmed the presence of diverse and specific microbial communities associated with different types of plastic which were different from microbes in the water column. Thus, microbial biofilms on the surface of plastic are called the “plastisphere” [78]. Additionally, omic studies are used to identify species responsible for the degradation of plastics [81].

### 3.2. Metagenomics of Biofilms on Antifouling Coatings and Biocides

Metagenomics of biofilms on antifouling coatings opened up a new world for researchers and industries. First, the studies indicated that microbial communities contain many species of prokaryotic and eukaryotic organisms. Diverse microbial communities were observed during a 1-year study of microbial biofilms on 11 biocidal antifouling coatings in Oman waters [61]. Communities were dominated by alpha- and gamma-Proteobacteria, Cyanobacteria, and Flavobacteria. Similarly, clear differences between communities representing three different types of coatings were observed. The 454 pyrosequencing of 16S rRNA demonstrated spatiotemporal changes in microbial communities on different antifouling coatings in French waters [82].

Second, metagenomic studies enable the identification of biofouling species present on fouling management coatings. The 454 pyrosequencing of 16S rRNA was used to evaluate microbes in biofilms on different coatings in French coastal waters [82]. The study reports that the biocidal coating reduced the abundance of Bacteroidetes. Similarly, clear differences in microbial communities developed on ZnO nanorod coatings and copper-based coating on fishing nets were recorded using Illumina MiSeq technology in Oman coastal waters [83]. The 16S and 18S amplicon Illumina MiSeq sequencing were used to evaluate prokaryotic and eukaryotic communities on different substrata in New Zealand waters [84]. The authors found that the orientation of the substrate, and the presence of antifouling coatings, impacted the composition of microbial communities. Similarly, clear differences in prokaryotic and eukaryotic communities on antifouling coatings were detected in another study [62]. However, in a different study, only prokaryotic communities were different between toxic and non-toxic coatings on sea gliders [63]. This study reported that the majority of bacteria species (1158) were shared between different protected and un-protected surfaces (Figure 4). Figure 4 is a Venn diagram, which is a graphical representation of microbial community analysis. Additionally, principal component analysis and clustering algorithms can be employed. The highest number of unique operational taxonomic units (OTUs) were observed on biocidal antifouling paint, while the lowest OTUs were found in non-biocidal paint. In another study, marine biofilms on biocidal and non-biocidal antifouling coatings were studied using 16S amplicon MiSeq sequencing [60]. Genera *Loktanella*, *Sphingorhabdus*, and *Erythrobacter* dominated biocidal coatings, while *Portibacter* dominated the fouling release ones. For decades people working with roof shingle fouling focused on one cyanobacterium *Gloeocapsa* spp., and when dust samples from shingles were analyzed *Gloeocapsa* proved to be a minor biofouling component [35]. Shingle manufacturers interested in understanding their biofouling problem could take advantage of microbiome analysis and use it to develop an effective antifouling defense.

Metagenomics is a powerful tool to detect the toxicity of coatings because of differences between microbial communities developed on toxic and non-toxic substrates. This can be used in forensic studies to detect compounds that are not listed in the recipes for industrial products. Ward et al. [85] reported the results of microbial growth on seven different kinds of plastic preproduction pellets in seawater. The investigators report that biofilms on plastic pellets were initially different and remained different on pellets, especially on PVC (polyvinil chloride) which was different from all the rest of the microbiomes on pellets at every sample interval. After 70 days, there were four distinct microbiomes on the pellets, with the convergence of microbiomes on similar plastics. The authors found that groups of bacteria associated with toxic fouling management coatings were found on some plastics at all time intervals suggesting compounds leaching from the plastic had a role in biofilm community composition. The more we understand microbial communities using metagenomics the better we will be able to manage human health, ecosystem health, and food security and develop effective antifouling and antimicrobial defenses.

### 3.3. Environmental DNA (e-DNA)

One use of metagenomics for metazoans in the biofouling community is environmental DNA (e-DNA) [86]. The idea is that DNA in water from near a fouling community can be filtered and then one can selectively amplify particular genes, such as Cytochrome Oxidase I (COI) or 18S ribosomal DNA, to determine members of the fouling community including invasive and cryptic species [87]. Obtained sequences can be compared with publicly available data and groups of interest can be identified. One advantage of this approach is that one does not need to be an expert in systematics because there are extensive public databases containing systematic information that can be accessed and used.

Several groups worldwide have developed this approach to make lists of the species in their biofouling communities. For example, e-DNA was used to detect the invasive golden mussel *Limnoperna fortunei* in Chinese waters [87]. Another study used e-DNA from plastic litter to detect four invasive species [88]. However, there are challenges associated with this technique. The e-DNA technique is too new to answer questions about the presence or absence of particular species and estimate their abundance at the time of sampling. No one knows the e-DNA shedding rates for different kinds of organisms. The sequences of some biofouling species are either absent in the databases or primers (for COI and 18S) and this prevents distinguishing them from other similar species. The DNA distribution in water is patchy [89]. The methods of sampling and DNA extraction can have an impact on the results [90]. Finally, all e-DNA are not equal and in complex communities, some dominant species can camouflage the presence of rare species.

During the development of this review, we realized that there are a lot of important unknowns in the development and use of “omic” techniques in the environment. For example, the half-life of arginine carboxy-terminal signal peptides in marine environments is approximately 2 h. The half-life of e-DNA is in a similar range. However, though e-DNA might still be present, the pieces could be too short to be useful to be amplified as intact sequences of DNA hundreds of base pairs in length are required to identify organisms. More generally, the question of shedding rates and forms of shedding for different kinds of metazoans remains open. For example, do mollusks produce the same amount of e-DNA per gram of living animal as arthropods or polychaetes, or cnidaria? There is plenty to do to improve “omic” techniques in the future.

## 4. Transcriptomics

Transcriptomics is defined as “everything RNA” (Figure 2). Transcriptomics allows us to detect DNA sequences that are transcribed in the current moment and used by biofouling organisms. Like other “omics” approaches, it has its jargon, incredible strengths, understated weaknesses, and the explosive growth of science on a rapidly expanding technological frontier (Figure 3). In the RNA world, Deep Sequencing means the same thing as Next Generation Sequencing in the DNA world and refers to the output of a very powerful sequencing platform. Transcriptomics has its limitations. Compared to DNA, RNA is less stable, thus, extra precautions need to be taken. Experts estimate that RNA amounts to as little as 1% of the total can be identified [25]. The result is the equivalent of a fingerprint for a cell at one point in time, or a disease state, or an individual, or, in our case, a community. This field has rapidly discovered some new kinds of RNA that are important to cellular function and disease states. One percent seems like a fine level of precision until one realizes that most enzymes occur at much lower percentages. One must also remember that like all “omic” techniques, processing and poorly understood amplification bias blurs the final picture. Nevertheless, transcriptomics is revolutionizing our understanding of how cells, organisms, and diseases work.

In biofouling research, transcriptomics has been used to investigate genes expressed during larval metamorphosis, and genes transcribed in biofilms. Bacteria regulate their cellular behavior using chemical molecules [91] during the quorum sensing process. A signal molecule, like acyl homoserine lactone, plays a crucial role in controlling biofilm formation and toxin production. When the concentration of signal molecules in the environment reaches the threshold level, it leads to the expression of genes responsible for biofilm formation, compound production, virulence and others. The treatment of biofilms with a quorum sensing blocker (furanone) changed the expression of 61 genes [92]. These genes were responsible for quorum sensing signal production, flagellar assembly, and aspartate kinase. Subsequently, changes in biofilms resulted in lower larval settlement. 

The effect of climate change on larval development was studied using a transcriptomic approach [44,45,46]. The experiment showed that ocean acidification suppressed the immune response pathways of pacific oyster *Crassostrea gigas* larvae [93]. Transcriptomics can provide additional information about the mechanism of action of antifouling compounds. The effect of Di(1H-indol-3-yl)methane on the transcription of DNA genes of the bryozoan larvae *Bugula neritina* showed that this “environmentally friendly” compound downregulates steroid hormone biosynthesis genes, which could have long-term effects on the bryozoan [25]. Transcriptomes have been used in the analysis of genes responsible for adhesive proteins of barnacles [94] and mussels [95]. In most cases, transcriptomics is combined with proteomics for a more complete analysis of expressed genes and produced proteins [96].

## 5. Proteomics

Proteomics is the study of peptides and their associated proteins (Figure 1). Proteomics is based on the use of high-resolution mass spectrometry [97,98]. In the standard approach, a researcher isolates proteins from the organism or community, fractionates them, breaks the proteins into peptides with a pure enzyme (i.e., trypsin), and identifies the mass of resulting peptides (Figure 5). Finally, extensive databases (such as NCBI) that contain masses and amino acid sequences of known peptides and proteins are used to determine peptides and their origins. Free user-friendly analysis tools (such as Mascot https://www.matrixscience.com/ (accessed on 16 June 2023) and Scaffold-TM 5 https://www.proteomesoftware.com/products/scaffold-5/ (accessed on 16 June 2023)) enable a novice to understand what proteins were present in the mixture. The databases provide a rich context that enables one to deduce the families of proteins to which their peptides belong.

As with all the omics, there is a complex language that includes trivial names for analysis techniques, i.e., Mascot, Scaffold, etc., and very practical concepts like false discovery rate and the likelihood that your peptide fragments are from specific families of proteins.

In the context of evolution, biologists’ problem-solving skills increased because they can use principles of evolution, like relatedness, to help solve biological puzzles. Proteins and their evolutionary relations in series, like protein structure and function, provide insight into protein evolution and relatedness and help us understand the biological phenomena. The protein literature is incredibly vast, making it difficult for a researcher to know all of it. For example, there is a compilation of over 700 papers that describe different kinds of proteolytic enzymes [98]. However, modern databases of protein sequences provide powerful insight into protein structure and function. Moreover, pathways common to all metazoans provide particular insights.

Proteomics of biofouling organisms can be used to identify proteins expressed during metamorphosis or during inhibition of metamorphosis (Table 3). Most of the studies conducted are with dominant biofouling species. Data for other fouling species are less common. In contrast to terrestrial species whose proteomes are well studied, marine species receive less attention.

One of the first studies analyzed the protein expression of a bryozoan *Bugula neritina* and a barnacle *Balanus* (*=Amphibalanus*) *amphitrite* during larval metamorphosis [99]. Induction and inhibition of metamorphosis for both species changed the phosphorylation status of proteins. Similar results were observed for the phosphorylation of proteins of the polychaete *Hydroides elegans* [49]. More than 700 proteins were identified during the metamorphosis of *B. amphitrite* [100]. Another study with the bryozoan *B. neritina* reported that proteins involved in energy metabolism were downregulated while proteins responsible for transcription and translation were upregulated during the metamorphosis of this species [101].

Proteomics enabled the study of the effect of climate change on the proteins of marine organisms (Table 3). The effect of multiple stressors associated with climate change on the proteome of oyster larvae *Crassostrea gigas* was investigated [102]. Interpretation of the data suggested that climate change will significantly impact proteins expressed in larval stages. Similar results were found for another species of oyster *Crassostrea hongkongensis* [103]. Upregulated and downregulated proteins have been found during the exposure of oyster larvae *Saccostrea glomerata* to acidic conditions [104]. Exposure of *Mytilus* species to different salinities leads to changes in proteins involved in energy metabolism and scavenging of reactive oxygen species [105].

Some proteomics studies focus on adhesive proteins (Table 3). It was hypothesized that metazoan glue curing was related to other processes where proteins coagulated in water. Testing the hypothesis with barnacle glue provided a preliminary conclusion that barnacle glue curing is related to blood clotting [106]. However, many biologists, biochemists, and structural protein chemists have not accepted the hypothesis. Fourteen years later, further work with collaborators from other disciplines confirmed the relationship between adhesive glue and hemolymph proteins [107]. In the study, researchers have shown that barnacle glue curing involves an ancient pathway, the innate immune response, that neutralizes pathogens. The researchers found evidence that in addition to neutralizing pathogens, the innate immune response provided chemistries, like reactive oxygen species, involved oxidative crosslinking and the processes involved in reworking and calcifying the barnacle glue. This has changed our view of biological adhesives. In addition to complex structural protein self-assembly, the process also involves many enzymes, a variety of reactive species from several sources, and secondary processes that result in calcification and curing for weeks if not months [107]. Peptides generated during and after glue formation and glue curing mediate gregariousness in barnacles, oysters, and a large number of other processes that organize marine communities [107].

Proteomics has several weaknesses. Many proteins from marine organisms, including biofouling organisms, are not well studied. Thus, their identification is hard. Complete genomes of most marine organisms are not available, which makes the prediction of proteins difficult. Thus, the function of novel proteins is difficult to predict. Because of the presence of a large number of proteins it is difficult to identify functionally important proteins, especially enzymes existing in lower numbers in comparison with the dominant ones. This problem can be mediated by employing additional techniques, such as antigen–antibody reporting which amplifies signals with reporter enzymes. As with all new techniques, time will enable a better understanding and interpretation of the proteomics data.
ijms-24-10518-t003_Table 3Table 3Relevant proteomic studies of marine biofouling organisms.GroupSpeciesPropose of StudyReferencesProkaryotesCyanobacteriaEffect of hydrodynamics[108]mixed communitiesBiocorrosion[109]mixed communitiesEffect of contaminants[110]Arthropoda*Balanus amphitrite*Proteins during larval metamorphosis[100]*Balanus amphitrite*Larval response to AF compound[111]
*Balanus amphitrite*Glue proteins[107]Bryozoa*Bugula neritina*Proteins during larval metamorphosis[101]*Bugula neritina*Impact of AF compound[112]Polychaeta*Hydroides elegans*Proteins during larval metamorphosis[49]Mollusca*Crasosstrea gigas*Effect of climate change[102]*Crassostrea hongkongensis*Effect of climate change[103]*Saccostrea glomerata*Effect of climate change[104]*Mytilus trossulus**M. galloprovincialis*Effect of salinity[105]AF—antifouling.


## 6. Metabolomics

Metabolomics studies all the metabolites found in organisms and the environments around them [113]. In a metabolomic project, metabolites are first extracted via polar and non-polar solvents. Then, crude extracts are fractionated using chromatography techniques (gas or liquid chromatography) and individual compounds are identified using high-resolution mass spectrometry (MS). Finally, the compounds are identified by comparison of their retention times, fragmentation patterns, and m/z values by comparison with compounds in libraries. However, due to the limitations of the methods at this point, only small molecules can be analyzed. Metabolomics can be targeted or untargeted [114]. In the targeted method, researchers focus on the analysis of individual metabolites involved in a specific pathway. This method can be used for “fingerprint” analysis of samples. In untargeted metabolomics, thousands of metabolites from an organism or community are analyzed. This method is useful for the identification of biomarkers.

For researchers working with biofouling, metabolomics is intriguing but daunting. Metabolomics can be used as a snapshot to examine the physiological impacts of toxins or biocides on organisms or communities [47,48]. The effect of an antifouling compound on metabolites of *Vibrio* sp. was studied [48]. Researchers found that during treatment some metabolites were downregulated while others were upregulated. Overall, the number of metabolomic studies is limited in biofouling studies due to a lack of basic understanding of the metabolic pathways in biofouling animals.

Combinations of omics techniques can be useful. Biofouling was studied on protected and non-protected surfaces using 18S amplicon sequencing and metabolomic analysis [47]. The results demonstrated that the metabolomes of biofouling communities developed on artificial substrata in different sites were different from each other. Metabolites of biofouling organisms found on different antifouling paints were also different. A combination of metabolomics with metagenomics approaches showed that different metabolomes were associated with different biofouling phyla [47]. A study of metabolomics and proteomics of a marine bacterium *Pseudoltermonas liplytica* in planktonic and attached cultures showed that different metabolites were produced [115]. Proteomics showed major differences in peptidases, oxidases, and membrane proteins.

Metabolomics can be used to determine biofouling species, especially in the case of microbes. Identification of bacteria by their metabolites is currently used by a MALDI-biotyper. However, the biotyper is not very effective for the identification of marine bacteria due to limited databases. On the other hand, *Persicivirga* (*Nonlabens*) *mediterranea*, *Pseudoalteromonas lipolytica*, and *Shewanella* sp. were identified based on their metabolites using LC-MS [116]. An MS/MS database for tested species of marine bacteria allowed the investigators to distinguish them based on their metabolites.

The main difficulties with metabolomics, as with many omics techniques, are the absence of standardized methods for the extraction and analysis of metabolites and the difficulties in the quantification and identification of compounds. Lack of information on the metabolic processes of marine organisms limits the analysis of metabolites. For example, the major fouling species barnacles, *Amphibalanus* (*=Balanus*) *amphitrite*, polychaetes *Hydorides elegans*, and bryozoans *Bugula neritna* are missing in the metabolic pathway MetaCyc (https://metacyc.org/ (accessed on 16 June 2023)) and BioCyc databases (https://biocyc.org/ (accessed on 16 June 2023)). An additional difficulty is associated with the analysis of the metabolome of complex communities that contain many species. The combination of metabolomics with other omic techniques, for example, proteomics and metagenomics, can provide additional insight into biofouling processes. Standardization of extraction and analysis methods as well as the development of new analytical methods for the identification of compounds will help in the future development of metabolomics.

## 7. Conclusions and Future Outlook

Our review demonstrates that the proportion of biofouling publications that utilize “omics” techniques is quite low. However, this number is constantly increasing. “Omics” provide information about the composition of biofouling communities and allow the detection of biofouling and invasive species. “Omics” approaches allow us to investigate the response of biofouling species to environmental factors and biocides, and opens new paths to study and better understand biofouling species and communities and their impact on marine installations. However, they have certain limitations that need to be considered during investigations (Figure 6). Some of these limitations are specific to each technique but others are general. Known limitations and possible solutions were highlighted during this review.

Combining “omics” approaches can yield significant advantages. Most combination approaches are used by multidisciplinary teams that investigates specific questions and have an interest in crossing the boundaries of each “omics” approach. Researchers in the “omics” silos have plenty of development to do within each technology and may find research at the boundaries less interesting.

Each “omics” technology has its strengths and weaknesses. All of them are limited by competition between targets, limits to the technologies, inherent biases in technological approaches, and lack of a larger biological perspective. Combining the techniques enables using one “omics” to overcome a shortcoming of another. As one can surmise, making this experimental stride requires a substantial understanding of the techniques involved and the ability to communicate across sub-disciplinary boundaries in the context of the biological questions asked.

Specialized language makes it difficult to understand and interpret the ‘omics’ data for biologists and non-specialists (Figure 6). Thus, help from colleagues and specialized agencies is essential. One solution is to simply look only for targeted genes or compounds or metabolic pathways. Large massive data require careful analysis and quality assessment. Additionally, researchers need to include informaticians or big data analytics in their research design. Another limitation is that biological meaning can be lost during the analysis.

Using “omic” approaches could be beneficial in the development of non-toxic or low-toxic antifouling defense. The toxicity of antifouling compounds and their mechanism of action can be tested using transcriptomics, metabolomics, and proteomics approaches. The presence of particular micro- and macro-fouling organisms on the surface of antifouling paints can be detected via metagenomic approaches. Additionally, one would be able to identify the role of microorganisms present on coatings or marine installations. Using these approaches will be easier and cheaper in the future. We believe that in the future even non-specialists would be able to identify the presence of invasive or biofouling species in industrial marine applications.

During the preparation of this review, we realized that most “omics” publications dealt with dominant species of micro- and macroorganisms. The information about other species, their genomes, proteins, and metabolites are limited in the databases. For some biofouling species, we have no information about their genes, proteins, and metabolic pathways (Figure 6). For example, even the very common *Balanus* genus has only two sequences in the NCBI database. Only one complete shotgun genome sequence for *Amphibalanus* (*=Balanus*) *amphitrite* was found (https://www.ncbi.nlm.nih.gov/nuccore/VIIS01000292.1 (accessed on 16 June 2023)). This makes it difficult to interpret the results, even for the dominant species. However, the data show that biofouling communities are highly diverse and composed of many biofouling species. At this point, it is difficult to use “omics” approaches for biofouling communities. Thus, in the future, more studies should be done to solve this and other problems associated with the use of “omics” technologies.

## Figures and Tables

**Figure 1 ijms-24-10518-f001:**
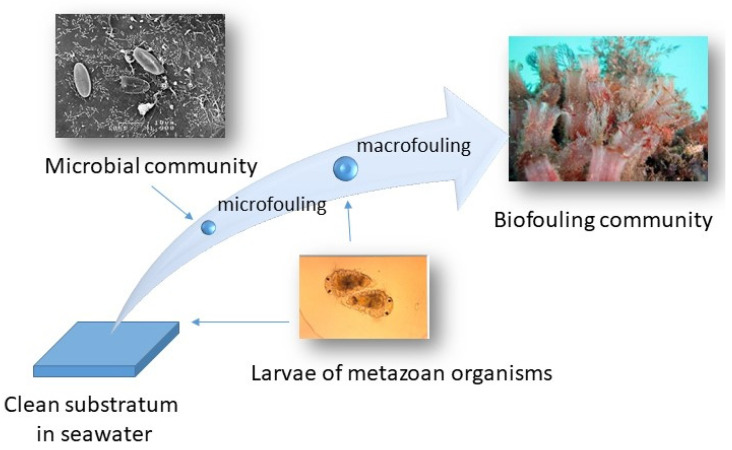
Schematic picture showing formation of micro- and macrofouling communities.

**Figure 2 ijms-24-10518-f002:**
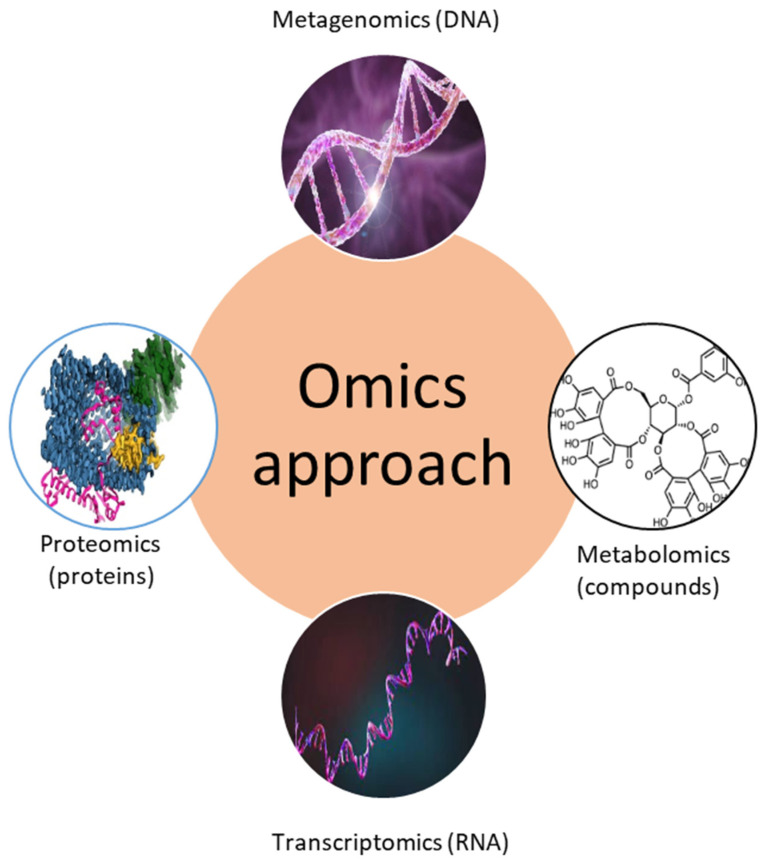
Different “omics” approaches to study biofouling. There are many more kinds of “omics” techniques, but these are not yet represented in biofouling studies.

**Figure 3 ijms-24-10518-f003:**
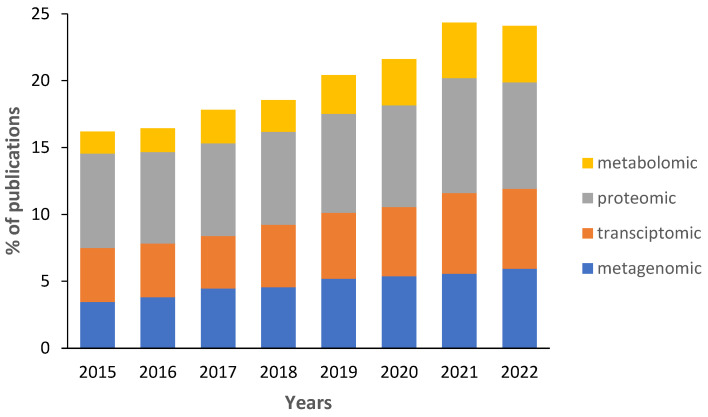
The percentage of biofouling publications that use metabolomic, proteomic, transcriptomic, and metagenomic methods. The data uses Google Scholar publications from 2015 to 2022. During the search, we used the keywords “biofouling” and “omics” terms (metabolomic, transcriptomic, proteomic, and metagenomic). Some of the studies used a combined approach which is not reflected by this graph.

**Figure 4 ijms-24-10518-f004:**
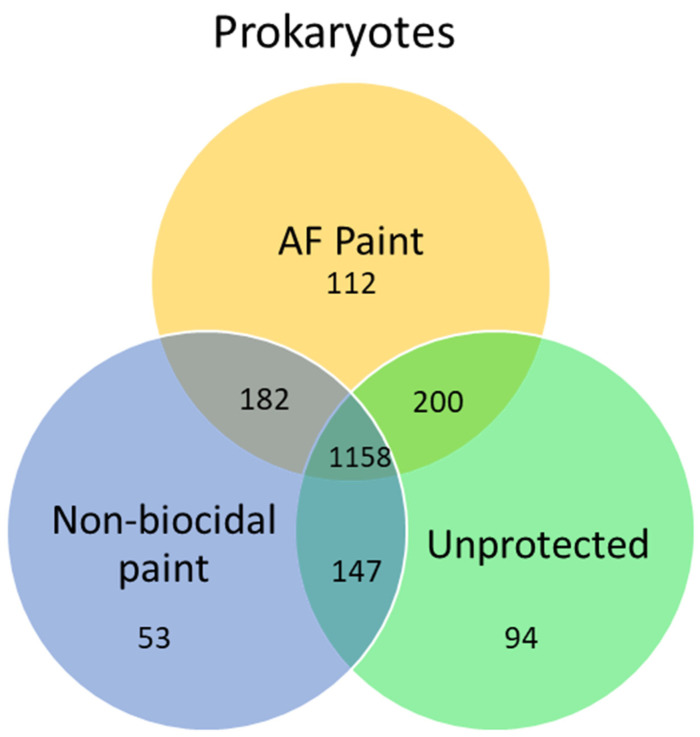
Venn diagram showing the number of shared and unique OTUs in prokaryotic communities studied via MiSeq Illumina and developed on sea gliders during the experiment of Dobretsov et al. [63] AF is a copper-based antifouling paint. Non-biocidal paint is a chitosan-based paint. Unprotected are not painted parts of the glider.

**Figure 5 ijms-24-10518-f005:**
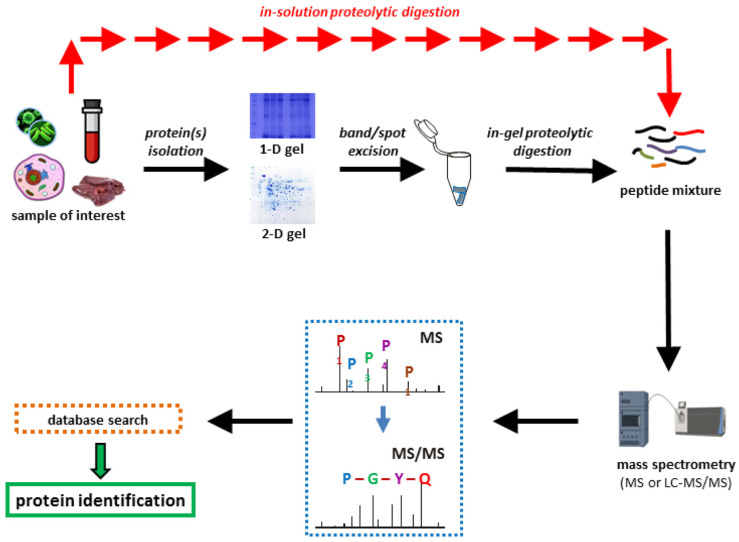
The workflow of proteomics. Reprinted with permission from [97]. 2020, MDPI.

**Figure 6 ijms-24-10518-f006:**
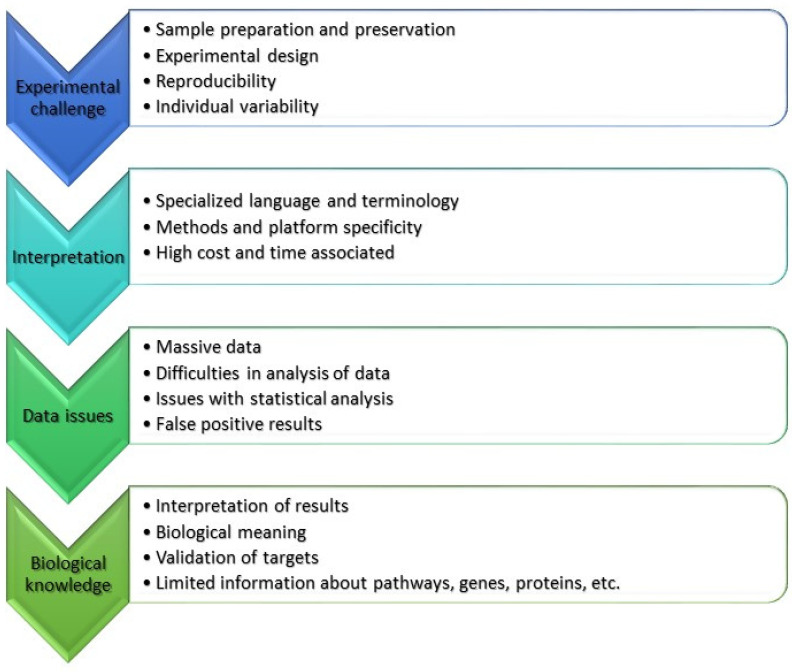
Limitations and challenges of “omics” techniques described in this review.

## Data Availability

Data are available upon request from the authors.

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
