# Peer review of "“Omics” Techniques Used in Marine Biofouling Studies"

_ijms, 2023, doi:10.3390/ijms241310518_

Round 1

Reviewer 1 Report

Title: Omics” techniques used in marine biofouling studies

The manuscript by Doritos and Rittschof et al. is notewrthy and requires revision before its publication in the IJMS as follows:

Comments

1.     In biofouling research, transcriptomics ….biofilms…..quorum sensing blocker (furanone) …… responsible for quorum sensing signal production…..a lower larval settlement. A signal molecule plays a key role in controlling biofilm, please add molecular mechanism and with few details i.e., Molecules 2022, 27(21), 7584. In addition, the role of monoclonal antibodies in targeting the Biofilm can also be provided in this section, i.e. Biomedicines 2023, 11(3), 765.

2.     The quantitative data, and suitable explanations with perspectives information’s are missing in most of sections. Please minor polished.

3.     Please add a few illustrations, to highlight the limitations and advantages of various “omics” techniques in the various applications. 

Minor changes are required.

Author Response

Comment 1.     In biofouling research, transcriptomics ….biofilms…..quorum sensing blocker (furanone) …… responsible for quorum sensing signal production…..a lower larval settlement. A signal molecule plays a key role in controlling biofilm, please add molecular mechanism and with few details i.e., Molecules 2022, 27(21), 7584. In addition, the role of monoclonal antibodies in targeting the Biofilm can also be provided in this section, i.e. Biomedicines 2023, 11(3), 765.

Reply: We included this information and references. See p.11. The text now reads: Bacteria regulate their cellular behavior using chemical molecules during quorum the sensing process. A signal molecule, like acyl homoserine lactone, plays a crucial role in controlling biofilm formation and toxin production. When the concentration of signal molecules in the environment reaches the threshold level, it leads to the expression of genes responsible for biofilm formation, compound production, virulence and others.  However, we did not include information on monoclonal antibodies, as this is out of scope of this review, which is about omics approaches.

  1. The quantitative data, and suitable explanations with perspectives information’s are missing in most of sections. Please minor polished.

 Reply: Since this is a review we cannot include a lot of information. Thus, it is not possible to include many quantitative data and discussions. However, we tried to include these as much as possible. The references should help the interested reader with interpretation.  With the help of reviewer 2 we did significant polishing and clarification.

  1. Please add a few illustrations, to highlight the limitations and advantages of various “omics” techniques in the various applications. 

Reply: Thank you for your suggestions. As suggested by the reviewer 1, we included figure 1 that shows formation of micro- and macrofouling on a submerged substratum. Additionally, we included figure 6 to highlight the limitations of various “omics” techniques.

Reviewer 2 Report

as attaced file.

Author Response

Reviewer 2 commented in a Pdf file.

Reply: All comments were accepted and the manuscript was revised accordingly.

Reviewer 2 made many detailed comments that greatly improved the readability and clarity of the presentation.   After making reviewer 2 corrections we edited the entire manuscript in the spirit of those comments.

Reviewer 3 Report

Dear Authors,

I read with great pleasure the article entitled "Omics" techniques used in marine biofouling studies, and I am aware that it is not an easy task to present the information as briefly as possible without excluding so many details about such a complex subject. The review article can represent valuable scientific material for all those who want to enhance their advanced knowledge about the formation and development of biofouling, as well as the current stage of quantifying specific dynamic processes (mostly versatile, directly referring to the fact that the fouling community emerges and adapts very rapidly in diverse environments) clear to biofouling.

To improve the quality of the article, I recommend the following:

  1. Add an image depicting the life cycle of biofouling in the Introduction section.
  2. Provide a brief overview of the different criteria for surface selection in biofilm formation. If there is any preference or indicator related to the quality of surfaces that other microorganism species adhere to.
  3. Including information on the repeatability or reproducibility of the analytical methods (such as mass spectrometry, infrared spectroscopy, chromatographic techniques coupled with mass spectrometry, etc.) used in biofouling evaluation. Is NMR analysis utilized in the assessment?
  4. Some studies employ statistical analysis methods or statistical models—for example, PCA (Principal Component Analysis) or clustering algorithms.
  5. Add representative citations to Table 1.
  6. Replace the links on pages 14 and 15 with corresponding (number) references from the reference list.

Author Response

Dear Authors,

I read with great pleasure the article entitled "Omics" techniques used in marine biofouling studies, and I am aware that it is not an easy task to present the information as briefly as possible without excluding so many details about such a complex subject. The review article can represent valuable scientific material for all those who want to enhance their advanced knowledge about the formation and development of biofouling, as well as the current stage of quantifying specific dynamic processes (mostly versatile, directly referring to the fact that the fouling community emerges and adapts very rapidly in diverse environments) clear to biofouling.

Reply: Thank you for your kind words, your context and highly valuable opinion. Our aim is exactly as you state.

To improve the quality of the article, I recommend the following:

Comment 1: Add an image depicting the life cycle of biofouling in the Introduction section.

Reply: It is difficult to provide a picture of the life cycle based on a brief explanation in this review. However, we tried our best to address this comment and produced figure 1 formation of micro and macrofouling on a submerged substratum.

Comment 2: Provide a brief overview of the different criteria for surface selection in biofilm formation. If there is any preference or indicator related to the quality of surfaces that other microorganism species adhere to.

Reply: Thank you for your comment. It is difficult to provide the different criteria for surface selection and biofilm formation. There are many variations and description would be a topic of another review. However, in order to address your comment, we include information that different microbes can attach to different substrata passively and actively. The text reads: Microfouling results from the active or passive movement of microbes onto surfaces from the water column. The chemical and physical properties of the substratum as well as the presence of other microbes determine the adhesion and biofilm formation of different bacteria. (p.2).  There are as many alternatives for metazoans as there are surface energies and potential interactions with microbes. We hope that there is a sense of this that comes through in the prose and in the added citations.

Comment 3. Including information on the repeatability or reproducibility of the analytical methods (such as mass spectrometry, infrared spectroscopy, chromatographic techniques coupled with mass spectrometry, etc.) used in biofouling evaluation. Is NMR analysis utilized in the assessment?

Reply:

The analytical methods, mass spectrometry, infrared spectroscopy, chromatographic techniques, various platforms are extremely reliable.  Variations observed are due to other aspects of the biological preparations such as the time interval for processing an unprotected sample and unknowns in complex biological mixtures.  Controls are essential for interpretation of the data output.  These need to be customized by experimentors for their specific experiments and the standard “omic” controls are part of the existing commercial  packages.

Comment 4: Some studies employ statistical analysis methods or statistical models—for example, PCA (Principal Component Analysis) or clustering algorithms.

Reply: Thank you for your comment. Unfortunately, the page limit does not allow use to include a section on statistical analysis of the “omics” data. However, we mentioned some of the methods in the text. The text reads: Finally, the data are analyzed by different statistical methods. (p.6). This is a Venn diagram, which is one of the ways of graphical analyses of microbial community analysis. Additionally, principal component analysis and clustering algorithms can be employed. (p.9)

Comment 5: Add representative citations to Table 1.

Reply: We added some citations to Table 1.

Replace the links on pages 14 and 15 with corresponding (number) references from the reference list.

Reply: It is not clear which links the reviewer mentioned. Are these the links to the databases? https://www.ncbi.nlm.nih.gov/nuccore/VIIS01000292.1, MetaCyc (https://metacyc.org/) and BioCyc databases (https://biocyc.org/). These are not references and cannot be included there.

Reviewer 4 Report

The authors have presented a review on the different ‘omics’ approaches- metagenomics, proteomics, metabolomics and transcriptomics to study the effect on biofouling. This review is very interesting and would certainly help in understanding the mechanism of biofouling from a genomics perspective. I recommend publication of it in its current form.

Author Response

Thank you for your kind words, your context and highly valuable opinion.

Round 2

Reviewer 1 Report

Accept as is

Reviewer 2 Report

Good Done, Congratulation